# Relationship between Corticospinal Excitability While Gazing at the Mirror and Motor Imagery Ability

**DOI:** 10.3390/brainsci13030463

**Published:** 2023-03-09

**Authors:** Jun Iwanami, Hitoshi Mutai, Akira Sagari, Masaaki Sato, Masayoshi Kobayashi

**Affiliations:** Division of Occupational Therapy, School of Health Science, Faculty of Medicine, Shinshu University, Matsumoto 390-8621, Japan

**Keywords:** motor imagery ability, motor evoked potential, transcranial magnetic stimulation, gazing at the mirror, mirror therapy

## Abstract

Mirror therapy (MT) helps stroke survivors recover motor function. Previous studies have reported that an individual’s motor imagery ability is related to the areas of brain activity during motor imagery and the effectiveness of motor imagery training. However, the relationship between MT and motor imagery ability and between corticospinal tract excitability during mirror gazing, an important component of MT, and motor imagery ability is unclear. This study determined whether the motor-evoked potential (MEP) amplitude while gazing at the mirror relates to participants’ motor imagery abilities. Twenty-four healthy right-handed adults (seven males) were recruited. Transcranial magnetic stimulation was performed while gazing at the mirror, and MEP of the first dorsal interosseous muscle of the right hand were measured. Motor imagery ability was measured using the Kinesthetic and Visual Imagery Questionnaire (KVIQ), which assesses the vividness of motor imagery ability. Additionally, a mental chronometry (MC) task was used to assess time aspects. The results showed a significant moderate correlation between changes in MEP amplitude values while gazing at the mirror, as compared with resting conditions, and assessment scores of KVIQ. This study shows that corticospinal excitability because of mirror gazing may be related to the vividness of motor imagery ability.

## 1. Introduction

The demand for rehabilitation for upper limb dysfunction is high because 70% to 80% of stroke survivors have limited upper limb functionality [1]. Several rehabilitation treatments for restoring motor functions and improving activities of daily living in patients with stroke exist [2]. In addition to traditional methods such as muscle strengthening, constraint-induced movement therapy (CIMT), electrical therapy, and active observation using mirror therapy, rehabilitation using new technologies such as brain-computer interface (BMI), virtual reality, and wearable devices have recently been developed [3,4,5]. Mirror therapy (MT) is considered an inexpensive and convenient approach compared with other treatments [3].

MT requires placing a mirror between the upper limbs, with the unaffected limb positioned in front of the mirror and the affected limb hidden behind. The patient observes the reflection of the moving, unaffected limb on the mirror, feeling that the affected limb is moving. MT is considered to change the neural circuits involved in motor control by integrating visual feedback and motor representations of the affected limb. MT could stimulate the primary motor, sensorimotor, and premotor cortex or mirror neurons, which is known to enhance motor function recovery of the upper limb in patients with stroke [6,7,8]. A meta-analysis review by Coquelin reported that MT improved motor function and activities of daily living in limbs affected by stroke [9]. However, the conditions and adaptations that enhance the effect of MT have not been established.

MT is closely related to behavioral observation and motor imagery. Previous studies using electroencephalography (EEG) [10] and somatosensory evoked potentials (SEPs) [11] have been conducted on the observation of movement through mirrors in MT, reporting activity in sensory-related brain regions. From these studies, it is thought that in MT, visual feedback from motion observation using a mirror produces both the illusion of kinesthetic sensation and the visual illusion and activates the perception-motion loop [10]. Motor imagery is defined as “a rehearsal of a movement that is reproduced in the brain’s working memory without actual sensory input or motor output” [12,13]. Motor imagery is a cognitive process related to motor control, such as planning, executing, and monitoring movement, and motor representations are generated in the brain during motor imagery. During motor imagery, neural circuits similar to those of actual movements, such as the primary motor cortex, premotor cortex, and supplementary motor cortex, are activated [14]. Motor imagery has traditionally been utilized in sports and is now being employed in rehabilitation, where it has demonstrated efficacy in enhancing the function of paralyzed hands and gait [15,16].

There are individual differences in motor imagery ability, and the vividness characteristics and time aspects of the image can be used to evaluate this. When the movement task is imaged without the actual motion, the vividness of a motor image is evaluated. The vividness of a motor image is classified as a visual image when it seems as if the participant is observing another person exercising and as a kinesthetic image when it seems as if the participant is exercising. Mental chronometry (MC) is a method of evaluating the time aspects of motor imagery that evaluates the difference between the imagined and actual times required for the execution of a given movement, in which smaller differences indicate higher motor imagery ability [17,18]. Another method to evaluate motor imagery ability is event-related desynchronization (ERD), which uses changes in EEG frequency components [19]. It is generally recognized that ERD occurs during exercise and motor imagery. However, this method requires special measurement equipment and is difficult to use easily in a clinical scene.

The relationship between motor imagery ability assessed by these methods and brain activity during motor imagery has also been clarified. It has been reported that there is a positive correlation between activity in the visual cortex during imagery and image clarity about individual differences in imagery ability and characteristic brain activation patterns during imagery [20]. Furthermore, Guillot et al. reported differences in the neural networks activated during motor imagery between people with high and low motor imagery ability [21]. Similarly, significant differences in brain activity patterns during gait imagery have been reported between groups with high and low motor imagery ability [22]. Previous studies showed a correlation between Motor Evoked Potential (MEP) during motor imagery and motor imagery ability [23,24]. The relationship between the effects of motor imagery and motor imagery ability has also been reported in healthy participants [25]. A recent study also reported that people with higher motor imagery ability had a better transfer of aftereffects of motor imagery tasks [26].

The above results show that individual differences in motor imagery may influence brain activation and its effects during tasks using motor imagery. However, the relationship between mirror therapy and motor imagery ability is unclear. It may therefore be important to consider this factor when considering the application of mirror therapy in future studies. MT consists of movement by the healthy side and observation of the mirror. Previous studies have reported that gazing at the mirror is an important factor in increasing the excitability of the spinal tract because of mirror therapy [27]. Therefore, this study focused on mirror gazing, an important element of mirror therapy. This study determined whether the brain activity produced by observing mirror images, one of the elements of mirror therapy, is related to motor imagery ability. This study investigated whether changes in MEP amplitude during mirror gazing compared to resting conditions were related to participants’ motor imagery ability.

## 2. Materials and Methods

### 2.1. Participants and Experimental Procedures

Twenty-four healthy young adults were recruited with a mean age of 21.7 years (standard deviation [SD] = 1.0; 7 males). All participants were right-handed, as determined using the Edinburgh inventory. In addition, participants were screened to ensure that they had no current or past neurological conditions, were not pregnant, and had no implants that may be affected by Transcranial magnetic stimulation (TMS).

Participants completed two types of motor imagery assessments. After completing the assessments, the participants rested enough before the MEP measurements were taken. This study measured MEP under two conditions: resting and gazing at the mirror. The motor task of gazing at the mirror involved abduction and adduction of the left index finger at a frequency of 0.5 Hz. The muscle from which MEP was derived was the right first dorsal interosseous (FDI), selected to correspond with the agonist muscle involved in the motor task performed by the opposite hand. MEP was measured during abduction exercises to ensure synchronization of MEP measurement with the motor tasks performed during gazing at the mirror. The gazing at the mirror condition measurement took approximately several minutes to complete.

### 2.2. Evaluation of Motor Imagery Ability

#### 2.2.1. Vividness of Motor Imagery Ability

We used the Kinesthetic and Visual Imagery Questionnaire (KVIQ) [28] to assess the vividness of motor imagery ability. The KVIQ requires participants to imagine performing 10 movements using (1) visual (watching yourself perform from an external viewpoint) and (2) kinesthetic (focus on feeling yourself perform the movement) imagery. Participants rated the vividness of each image using a 5-point Likert scale, with scores from each item summed to provide a vividness score for each component between 10 and 50 (with lower scores indicating more vivid images).

#### 2.2.2. Time Aspects of Motor Imagery Ability

We used a MC task to assess the time aspect of motor imagery ability. The upper extremity MC task did not include a predetermined motor task. Rather, we selected motor tasks based on the criteria that participants were proficient in the task and that each component of upper extremity motor function (i.e., pinching, reaching, and releasing) was incorporated. Specifically, for this study, we chose the tenth task (i.e., picking a pin and placing it into a hole in front of the pin six times) of the Simple Test for Evaluating Hand Function (STEF), which is used to evaluate upper limb motor function. First, the participants were asked to imagine themselves performing the exercise task and measuring time with a stopwatch. Participants were asked to perform the imagery task using kinesthetic imagery as fast and as realistic as possible. Next, participants were instructed to perform the same exercise task and measure the time. Finally, the absolute value of the difference between those times was taken as the MC score.

### 2.3. Evaluation of the Excitability of the Primary Motor Cortex

The TMS method was used to assess activity in the primary motor cortex. This non-invasive approach depolarizes brain neurons by utilizing a magnetic field and MEPs from peripheral muscles upon applying TMS to the primary motor cortex. TMS was performed using a magnetic stimulator (Magstim 200: The Magstim Company Ltd., Whitland, UK) with a 70-mm figure-of-eight coil. The TMS stimulation site was located around the left motor cortex (C3), as defined by the international 10–20 method. The coil was positioned so that the intracranial-induced current flowed from backward to forward. The coil was moved 1 cm for each participant to determine the lowest stimulation threshold (hot spot) necessary for measuring the MEP from the target muscle. In addition, the area where the largest MEP waveform from the right FDI muscle was identified.

After the stimulation site was determined, the scalp was marked. The TMS stimulation intensity was set to 1.2 times the resting threshold (the minimum motion threshold at which the MEP amplitude of 50 μV could be derived more than 5 times out of 10 times) for each participant. In the gazing at the mirror condition, a photoelectric sensor (FS-N10, KEYENCE Co., Ltd., Osaka, Japan) was installed in the mirror box, and the output signal was sent to the magnetic stimulator when the left index finger abduction movement blocked the infrared light from the sensor. The infrared light was set to be blocked when the index finger reached the maximum abduction range (Figure 1). The number of TMS stimulations was 15 for each condition. The mirror box was 300 mm long, 450 mm wide, and 200 mm high, with easy access to the inside of the box and an adjustable mirror angle. The participants sat in a stable chair and were instructed to place both upper limbs to the same depth in a mirror box placed on a desk in front of them. The angle of the desk and mirror was adjusted so that the participants could gaze into the mirror and see their left hand reflected as their right hand. The MEP from the right FDI was derived with a bipolar surface electrode (Blue Sensor NF, Ambu, Denmark).

The distance between the electrodes was 1 cm. The derived EMG signals were transmitted through an analog interface (FA-DL-720, 4Assist Inc., Tokyo, Japan) to a data recording and analysis system (Power Lab 8/30, AD Instruments Pty Ltd., Oxford, UK). The MEP waveforms were converted to A/D at a sampling frequency of 40 kHz using a data recording and analysis system (Power Lab 8/30, AD Instruments Pty Ltd., UK) and recorded on a personal computer together with the TMS signal and the output signal of the photoelectric sensor. The recorded MEP waveforms were processed with a 20–2000 Hz band-pass filter. MEP amplitude values (peak-to-peak) were measured from the recorded MEP waveforms for each condition. After measuring the amplitude values, the average MEP amplitude (MEP amplitude) of 13 waveforms was calculated, excluding the maximum and minimum values among the 15 MEP waveform amplitudes obtained for each condition. The resting amplitude values normalized the average MEP amplitude values for the gazing the mirror condition, and this was used as the MEP ratio.

### 2.4. Statistical Analyses

The Shapiro–Wilk test was used to evaluate each outcome’s normal distribution of the scores. KVIQ scores were calculated for visual and kinesthetic imagery, and between-group comparisons were made using a paired t-test. MC scores for the time required to perform the task on the images, the actual time required, and the absolute value of the difference between them were calculated. The two groups were compared using a paired t-test for the image and the time required. Spearman’s rank correlation coefficient tests were used to determine the relationship between the results of the motor imagery assessment and the MEP ratio. In all cases, the significance level was set at 5%. All statistical analyses were performed with EZR (Saitama Medical Center, Jichi Medical University, Saitama, Japan), a graphical user interface for R (The R Foundation for Statistical Computing, Vienna, Austria). More precisely, EZR is a modified version of R commander designed to add statistical functions frequently used in biostatistics [29].

### 2.5. Ethics Approval

This study was conducted according to the Declaration of Helsinki, and the Ethics Committee of Shinshu University (approval no. 4283) approved it. The study was registered at the University Hospital Medical Information Network Clinical Trial Registry (registration no. UMIN000037149). All participants provided written informed consent to participate in this study.

## 3. Results

The results for the motor imagery tasks are presented in Table 1. There was a significant difference between visual and kinesthetic imagery scores on the KVIQ (*p* = 0.00269). For MC, there was a significant difference between the time measured in the image and the actual execution time (*p* = 0.00101). More detailed results for motor imagery ability are presented in Appendix A.

The typical recorded MEP raw waveforms over the right FDI muscle belonging to a representative participant are shown in Figure 2. The average MEP amplitude values (mV) for each participant under resting and mirror conditions are shown in Table 2. The mean MEP amplitude value significantly differed between the conditions (*p* = 0.00000898). The MEP amplitude values and MEP ratios for individual participants are shown in Appendix A.

Figure 3 and Figure 4 shows a scatter plot of the correlation between the evaluation results of each motor imagery ability and the MEP ratio. There was a significant correlation between the KVIQ (visual imagery) and MEP ratio (r = 0.505, *p* = 0.0118). Additionally, there was a moderate correlation between the KVIQ (kinesthetic imagery) and MEP ratio (r = 0.521, *p* = 0.0091). There was no significant correlation between the MC score and MEP ratio (r = −0.372, *p* = 0.0738).

## 4. Discussion

The therapeutic effects of mirror therapy on stroke survivors have been reported in the subacute [30], recovery, and chronic [3] phases. A systematic review of mirror therapy has also reported a certain level of efficacy [9,31], but the mechanisms and indications have not been fully elucidated [7]. This study focused on mirror gazing, one of the important components of mirror therapy, and aimed to determine whether MEP amplitude during mirror gazing was correlated with motor imagery ability. The results showed a significant moderate correlation between the vividness of motor imagery by KVIQ and the MEP ratio by mirror gazing. There was no correlation between the time aspects of motor imagery by MC and the MEP ratio. This study shows that there may be a relationship between the motor cortex activation induced by mirror gazing and the vividness of motor imagery.

Previous studies on motor imagery ability have examined the relationship between Parkinson’s disease [32,33] and multiple sclerosis [34] and motor imagery ability. Additionally, various reports have shown that the neural networks activated during motor imagery differ depending on the motor imagery ability [21,22], that the excitability of corticospinal tracts during motor imagery correlates with motor imagery ability [23,24], and that motor imagery ability is related to the effectiveness of motor imagery training [25]. Although it has been reported that motor observation using mirrors is important in mirror therapy, no previous studies have examined the relationship between brain activity during mirror gazing and motor imagery ability. Therefore, this study is novel in clarifying the relationship between motor imagery ability and corticospinal tracts during mirror gazing.

Significant moderate correlations were found in this study between the respective scores of the KVIQ, which assesses the vividness of motor imagery, and the MEP ratio during mirror gazing, indicating that mirror gazing enhances the excitability of the corticospinal tracts in individuals with the higher vividness of motor imagery. Mirror gazing may have produced the illusion of movement or visual illusions and evoked motor imagery. Participants were also required to produce a kinesthetic or visual motor image when the vividness of the motor image was assessed. Similar neural networks may have been activated in both mirror gazing and vividness of motor imagery assessment, which may be relevant to these results. This result is similar to previous studies showing that corticospinal tract excitability during motor imagery is related to motor imagery ability [23,24], showing that motor imagery ability may be a potential factor influencing corticospinal tract activation by MT. It may also explain why previous reports of the effects of mirror therapy interventions have varied. In the future, it may be important to assess motor imagery ability when examining the effectiveness of mirror therapy.

However, no correlation was found between the MEP ratio and mental chronometry task scores while gazing at the mirror. Previous studies have reported that stroke survivors perform worse on mental chronometry tasks [35,36,37] and that multiple sclerosis is associated with mental chronometry while walking [38]. However, only one previous study examined the relationship between brain–machine interface adaptation and mental chronometry tasks [39]. This repots is the first of its kind because no study has examined the relationship with gazing at the mirror. This study’s results show no relationship between the effects of gazing at the mirror and motor imagery ability as measured by mental chronometry tasks. However, it has been reported that the results of mental chronometry tasks vary depending on the motor task used [40]. We should have used motor tasks similar to those performed in gazing at the mirror to examine the relationship between brain activity while gazing at the mirror and motor imagery ability using mental chronometry. In this study, we used a task that required pinching and reaching with the upper limbs, but this study did not require reaching movements, and the difference may have affected the results. Further investigation is needed to change the motor task in the mental chronometry task.

Scientific advances have led to the development of various rehabilitation techniques for stroke survivors with motor paralysis. However, it is important to note that the effectiveness of each method may vary from patient to patient. For example, rehabilitation using virtual reality is a method that uses visual information and may be related to the vividness of motor images used in this study. It is important to clarify its application and factors to determine which treatment is more effective for a patient in a clinical situation. Therefore, this may reduce the time and financial burden on the patient and consequently improve the quality of life. Further research is needed on the relationship between the effectiveness of rehabilitation and individual patient characteristics.

## 5. Limitations

This study has several limitations. First, there are other methods to evaluate motor imagery ability, such as the mental rotation of the hand task, in addition to the motor imagery vividness and mental chronometry tasks used in this study. There are also other methods to assess motor imagery vividness, such as the MIQ-RS. In addition, there is a more specific method of evaluating motor imagery using EEG. Further studies are needed to investigate the relationship between gazing at the mirror and other motor imagery abilities and assessment methods. Second, this study was conducted on healthy participants, and it is unclear whether similar results would be obtained in stroke patients.

Finally, the effectiveness of gazing at the mirror is evaluated by MEP. It is necessary to clarify the changes in MEP and the relationship between function and performance to determine its clinical application.

## 6. Conclusions

This study investigated the correlation between MEP ratio by mirror gazing and motor imagery ability. The results showed that changes in corticospinal tract excitability induced by mirror gazing correlated with vividness of motor imagery. This study did not conduct training as mirror therapy, nor did it examine the effects on performance, so it is impossible to clarify the adaptation of mirror therapy. Nevertheless, the possibility of a relationship between the excitability of corticospinal tracts because of mirror gazing, which is an important component of mirror therapy, and vividness of motor imagery was demonstrated, showing that it is important to include assessment of motor imagery ability in order to verify the effects of mirror therapy.

## Figures and Tables

**Figure 1 brainsci-13-00463-f001:**
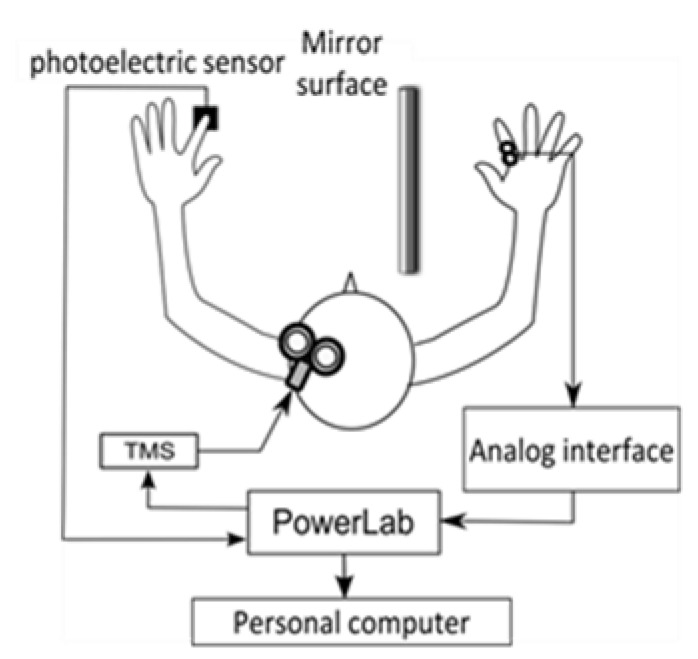
Experimental system.

**Figure 2 brainsci-13-00463-f002:**
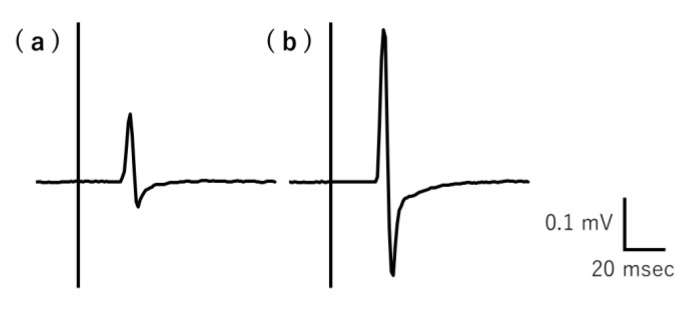
Typical MEP raw waveforms for resting and gazing at the mirror conditions recorded by a representative participant. The MEP amplitude was lower during the (**a**) rest condition than the (**b**) gazing at the mirror condition. MEP, motor-evoked potential.

**Figure 3 brainsci-13-00463-f003:**
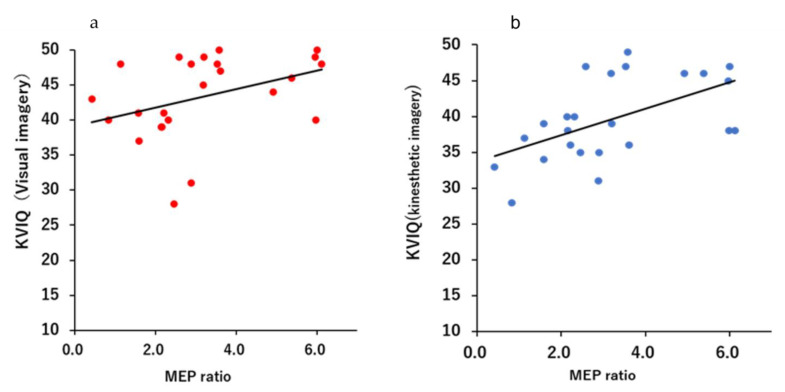
Scatterplots demonstrating correlations between the KVIQ score and the MEP ratio: (**a**) Relationship between the MEP ratio and visual imagery of the KVIQ; (**b**) Relationship between MEP ratio and kinesthetic imagery of the KVIQ.

**Figure 4 brainsci-13-00463-f004:**
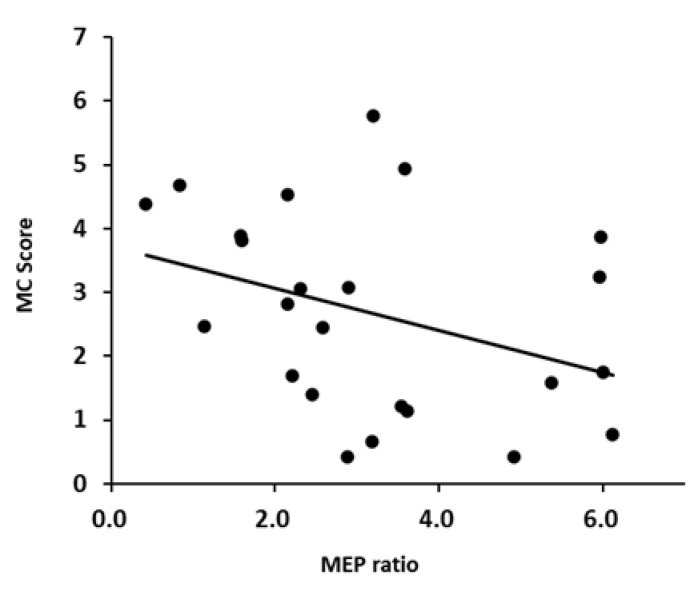
Scatterplots demonstrating correlations between the MC score and MEP.

**Table 1 brainsci-13-00463-t001:** Summary of motor imagery ability measures.

Measure	Component	Mean (SD)
KVIQ	Visual imagery Kinesthetic imagery	43.3 (5.8) 39.9 (5.7)
Mental chronometry (s)	Imaged time Actual time MC score	8.9 (3.4) 10.8 (1.9) 2.7 (1.5)

**Table 2 brainsci-13-00463-t002:** Summary of Mean of MEP amplitude (mV).

Condition	Mean (SD)
Rest condition	0.20 (0.11)
Gazing at the mirror condition	0.55 (0.35)

## Data Availability

The data used to support the findings of this study are presented in Appendix A.

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
