# Peer review of "Relationship between Corticospinal Excitability While Gazing at the Mirror and Motor Imagery Ability"

_brainsci, 2023, doi:10.3390/brainsci13030463_

Round 1

Reviewer 1 Report

Comments and Suggestions for Authors

This study was investigated that whether the motor imagery ability is related to MEP during the mirror therapy. As a results, MEP during the mirror therapy was related of vividness of motor imagery. However, this study has serious concern points.

1. In this study, theological background and hypothesize is not clear. Why did you think the motor imagery ability was associate to MEP during the mirror therapy? Would you please explain to me the theological background and hypothesize in this study.

2. It is unclear the method of Mirror therapy. How is task? What long time? Which limb?

3. Author used MC of STEF as motor imagery ability assessment. Why did you use this task?

  What is the relationship between this task and MEPs during mirror therapy?

I would consider using an MC task which is similar to the mirror therapy task.

Author Response

Response to Reviewer 1 Comments

Thank you very much for reviewing our manuscript and offering valuable advice.

We have addressed your comments with point-by-point responses, and revised the manuscript accordingly.

Point 1: In this study, theological background and hypothesize is not clear. Why did you think the motor imagery ability was associate to MEP during the mirror therapy? Would you please explain to me the theological background and hypothesize in this study.

Response 1: We thank the reviewer for these insightful comments. We have made significant revisions to the introduction to make the background and hypotheses clearer. The revised sections are lines 45-53, 62-73, and 76-79.

(lines 45-53: Mirror therapy is closely related to motor imagery and behavioral observation. Motor imagery is defined as "a rehearsal of a movement that is reproduced in the brain's working memory without actual sensory input or motor output“ [11,12]. Motor imagery has traditionally been utilized in sports and is now being employed in rehabilitation as well, where it has demonstrated efficacy in enhancing the function of paralyzed hands and gait [13,14]. Previous studies with healthy subjects have reported an association between the effects of imagery training and motor imagery ability [15]. However, the relationship between motor imagery ability and mirror therapy that uses motor imagery is not clarified.)

(lines 62-73: Another method to evaluate motor imagery ability is event-related desynchronization, which uses changes in electroencephalography (EEG) frequency components [18]. It is generally recognized that event-related desynchronization (ERD) occurs during exercise and motor imagery; however, this method requires special measurement equipment and is difficult to use easily in a clinical scene.

Transcranial magnetic stimulation (TMS) is another method employed to elucidate brain activity during motor imagery. This non-invasive approach depolarizes brain neurons by utilizing a magnetic field and measures Motor Evoked Potentials (MEPs) from peripheral muscles upon applying TMS to the primary motor cortex. Previous studies on brain activity during motor imagery using TMS have reported increased corticospinal tract excitability during motor imagery and movement observation [19,20]. In addition, previous research demonstrates that MEP amplitude increases during mirror therapy [21].)

(lines 76-79.: We hypothesized that brain activation by mirror therapy would be correlated with motor imagery ability, based on previous studies showing that motor imagery training and brain activation are related to motor imagery ability.)

Point 2: It is unclear the method of Mirror therapy. How is task? What long time? Which limb?

Response 2: Thank you for your important suggestion. I have added a lot of information about the method used in the mirror therapy, the content of the task, the duration of the task, and which limbs were used. The added part is line 94-101.

(line 94-101: In this study, MEP was measured under two conditions: resting and MT. The motor task of MT involved abduction and adduction of the left index finger at a frequency of 0.5 Hz. The muscle from which MEP was derived was the right first dorsal interosseous (FDI), which was selected to correspond with the agonist muscle involved in the motor task performed by the opposite hand. MEP was measured during abduction exercises to ensure synchronization of MEP measurement with the motor tasks performed during mirror therapy. The MT condition measurement took approximately several minutes to complete.)

Point 3: Author used MC of STEF as motor imagery ability assessment. Why did you use this task?

What is the relationship between this task and MEPs during mirror therapy?

I would consider using an MC task which is similar to the mirror therapy task.

Response 3: Thank you for pointing out the inadequacies of the manuscript.

As you pointed out, the reason for using STEF for MC was not stated. We have added this reason in the Methods (lines 113-118).

(Line 113-118: The upper extremity mental chronometry task did not include a predetermined motor task. Rather, we selected motor tasks based on the criteria that subjects were proficient in the task and that each component of upper extremity motor function (i.e., pinching, reaching, and releasing) was incorporated. Specifically, for this study, we chose the tenth task (i.e., picking a pin and placing it into a hole in front of the pin six times) of the Simple Test for Evaluating Hand Function (STEF), which is used to evaluate upper limb motor function.)

Also, as the reviewer pointed out, I added a note in the discussion (line 246-249) that the MC task should have been similar to the miller therapy task.

(line 246-249: To examine the relationship between brain activity during mirror therapy and motor imagery ability using mental chronometry, perhaps we should have used motor tasks similar to those performed in mirror therapy.)

Reviewer 2 Report

Comments and Suggestions for Authors

The authors presented a well done research on the determination whether motor imagery ability relates to cortiscospinal  excitability while performing mirror therapy. They argued that changes in motor imagery evoked potential amplitude values due to mirror therapy against resting conditions, were significantly associated with assessment results vividness of motor imaginary ability. Authors go on hypothesizing that it may exist a relation between the mirror therapy and the vividness of motor imagery ability.

The background underlying this study and the data gathering process are well presented, alongside an appropriate exposition of the materials and methods related with the research. I’m having, however, few observations that authors should consider while improving this paper as outlined below.

·         Line 56 states that “the purpose of the study was to clarify the adaptation of MT”. I don’t see how this purpose relates back to the one in line 11. Please clarify.

·         Following the previous point, line 18 states a significant association between MEP amplitude changes and assessment results. The statistical tests presented don’t assess association, only differences and correlations. Please clarify.

·         From figures 2 and 3, correlations look very scattered. Hence, it is not safe arguing about the existence of any correlation, even if the significant test says so.

Author Response

Response to Reviewer 2 Comments

Thank you very much for reviewing our manuscript and offering valuable advice.

We have addressed your comments with point-by-point responses, and revised the manuscript accordingly.

Point 1: Line 56 states that “the purpose of the study was to clarify the adaptation of MT”. I don’t see how this purpose relates back to the one in line 11. Please clarify.

Response 1: We thank the reviewer for the careful review of the manuscript. The purpose stated in the abstract and introduction has been revised to be clearer. The revised parts are lines 11-12 and 80-84.

Point 2: Following the previous point, line 18 states a significant association between MEP amplitude changes and assessment results. The statistical tests presented don’t assess association, only differences and correlations. Please clarify.

Response 2: We appreciate the reviewer’s suggestions. In response to a reviewer's suggestion, we have corrected lines 17-18.

Point 3: From figures 2 and 3, correlations look very scattered. Hence, it is not safe arguing about the existence of any correlation, even if the significant test says so.

Response 3: Thank you for your valuable suggestions. Since the correlation coefficient exceeded 0.5 in the current statistical analysis, we considered that there was a correlation between motor imagery ability and changes in MEP amplitude values during mirror therapy.

However, as the reviewer pointed out, the current version emphasizes only a significant correlation; therefore, we have revised the wording to explain that there is a significant moderate correlation. In addition, we have also changed the wording to indicate that there may be a relationship between motor imagery ability and corticospinal tracts during mirror therapy. The revised lines are 213-216 and 224.

(line 213-216: The results showed a significant moderate correlation between motor imagery vividness using the KVIQ and the MEP ratio during mirror therapy. This suggests that there may be a relationship between the vividness of motor imagery and the activation of the motor cortex induced by mirror therapy.)

Reviewer 3 Report

Comments and Suggestions for Authors

 Thank you for the interesting work. The article is well structured, the used language is adequate, however, I have some remarks and suggestions:

Lines 28-29

Several rehabilitation treatments for restoring motor functions and improving activities of daily living in patients with stroke exist

I invite authors to expand the literature review on the uses of new technologies, such as brain-computer interfaces, EEG, and virtual reality for stroke rehabilitation (see 10.1088/1741-2552/aba162 10.1016/j.procs.2020.09.270 10.1016/j.compbiomed.2020.103843). Many of them use in the reality motor imagery paradigm, as your article takes into consideration the visual and kinaesthetic parts, it would be nice to see also this aspect later discussed in the discussion chapter.

Lines 47-49

Mental chronometry is a subset of motor imagery that evaluates the difference between the imagined and actual times required for the execution of a given movement, in which smaller differences are indicative of higher motor imagery ability [8,9].

I would not call it a subset, but the technique that is used to evaluate one's ability to perform motor imagery. There are also other techniques, such as EEG Event-related (De) Synchronization, but these are more advance and require specific equipment.

Line 66

 participants took sufficient rest before the MPE measurements were taken.

MPE is undefined (a typographical error? -> MEP?)

Lines 87-89

The TMS stimulation site was the hand motor area of the left cerebral cortex (Cz) determined by the international 10-20 method, and the coil was moved with a 1-cm sense concerning the line connecting the eyebrow, the external occipital ridge, and the two external auditory foramina.

Cz corresponds to the central area and has a very precise location, I would expect it more on the positions of C1-C3-C5 to the left cerebral cortex. I also expect differences among subjects.

Line 90

The area where the MEP  waveform from the right FDI was the largest was identified.

FDI is not defined.

Line 122

KVIQ scores were calculated for visual and kinesthetic imagery, and between-group comparisons were made using a paired t-test.

I wonder if the authors test the normality of the data before applying the t-test.

Line 195

On the other hand, no correlation was found between the MEP ratio and mental chronometry task scores during Miller therapy.

Miller therapy undefined in the previous text

Line 203

However, it has been reported that the results of mental chronometry tasks vary depending on the motor task used

I agree it is not a perfect measure to evaluate one's ability to perform MI.

Line 210-211

This study has several limitations. First, there are other methods to evaluate motor  imagery ability, such as the mental rotation of the hand task, in addition to the motor  imagery vividness and mental chronometry tasks used in the present study

Yes, I agree, EEG for example is much more specific, at least for upper-limb evaluation of Motor Imagery.

Author Response

Response to Reviewer 3 Comments

Thank you very much for reviewing our manuscript and offering valuable advice.

We have addressed your comments with point-by-point responses, and revised the manuscript accordingly.

Point 1: Lines 28-29: Several rehabilitation treatments for restoring motor functions and improving activities of daily living in patients with stroke exist

⇒I invite authors to expand the literature review on the uses of new technologies, such as brain-computer interfaces, EEG, and virtual reality for stroke rehabilitation (see 10.1088/1741-2552/aba162 10.1016/j.procs.2020.09.270 10.1016/j.compbiomed.2020.103843). Many of them use in the reality motor imagery paradigm, as your article takes into consideration the visual and kinaesthetic parts, it would be nice to see also this aspect later discussed in the discussion chapter.

Response 1: I sincerely appreciate your advice to improve my manuscript. I have revised the Introduction and the Discussion after taking the reviewer's advice. The revised parts are lines 31-34 and lines 253-261.

(lines 31-34: In addition to traditional methods such as muscle strengthening, constraint-induced movement therapy (CIMT), electrical therapy, and active observation using mirror therapy, rehabilitation using new technologies such as brain-computer interface (BMI), virtual reality, and wearable devices have recently been developed [3-5].)

(lines 253-261: Scientific advances have led to the development of various rehabilitation techniques for stroke survivors with motor paralysis. However, it is important to note that the effectiveness of each method may vary from patient to patient. For example, rehabilitation using virtual reality is a method that uses visual information and may be related to the vividness of motor images used in this study. To determine which treatment is more effective for a patient in a clinical situation, it is important to clarify its application and factors. This may reduce the time and financial burden on the patient and consequently improve the quality of life. Further research is needed on the relationship between the effectiveness of rehabilitation and individual patient characteristics.)

Point 2: Lines 47-49: Mental chronometry is a subset of motor imagery that evaluates the difference between the imagined and actual times required for the execution of a given movement, in which smaller differences are indicative of higher motor imagery ability [8,9].

⇒I would not call it a subset, but the technique that is used to evaluate one's ability to perform motor imagery. There are also other techniques, such as EEG Event-related (De) Synchronization, but these are more advance and require specific equipment.

Response 2: Thank you for your important suggestion. I have revised the wording and added the description of EEG. The revised parts are line 59 and 62-66.

(Line 59 method)

(Lines 62-66: Another method to evaluate motor imagery ability is event-related desynchronization, which uses changes in electroencephalography (EEG) frequency components [18]. It is generally recognized that event-related desynchronization (ERD) occurs during exercise and motor imagery; however, this method requires special measurement equipment and is difficult to use easily in a clinical scene.)

Point 3: Line 66: participants took sufficient rest before the MPE measurements were taken.

⇒MPE is undefined (a typographical error? -> MEP?)

Response 3: We thank the reviewer for the careful review. It was a spelling mistake and I have corrected it. The revised part is line 93.

Point 4: Lines 87-89: The TMS stimulation site was the hand motor area of the left cerebral cortex (Cz) determined by the international 10-20 method, and the coil was moved with a 1-cm sense concerning the line connecting the eyebrow, the external occipital ridge, and the two external auditory foramina.

⇒Cz corresponds to the central area and has a very precise location, I would expect it more on the positions of C1-C3-C5 to the left cerebral cortex. I also expect differences among subjects.

Response 4: Thank you for your important suggestion. The description has been revised substantially. The revised part is line 125-129.

(Lines 125-129: The TMS stimulation site was located around the left motor cortex (C3), as defined by the international 10-20 method. The coil was positioned in such a way that the intracranial-induced current flowed from backward to forward. To determine the lowest stimulation threshold (hot spot) necessary for measuring the MEP from the target muscle, the coil was moved 1 cm for each subject.)

Point 5: Line 90: The area where the MEP  waveform from the right FDI was the largest was identified.

⇒FDI is not defined.

Response 5: Thank you for your suggestion.

The FDI appeared earlier because the manuscript was revised according to other reviewers' suggestions. Therefore, we defined it in line 96.

Point 6: Line 122: KVIQ scores were calculated for visual and kinesthetic imagery, and between-group comparisons were made using a paired t-test.

⇒I wonder if the authors test the normality of the data before applying the t-test.

Response 6: Thank you for your suggestion.

I actually performed the t-test after conducting the normality assessment. The text has been added to lines 161-162.

(lines 161-162: The Shapiro-Wilk test was used to evaluate the normal distribution of the scores for each outcome.)

Point 7: Line 195: On the other hand, no correlation was found between the MEP ratio and mental chronometry task scores during Miller therapy.

⇒Miller therapy undefined in the previous text

Response 7: We thank the reviewer for the careful review. It was a spelling mistake and I have corrected it. The revised part is line 238.

Point 8: Line 203: However, it has been reported that the results of mental chronometry tasks vary depending on the motor task used

⇒I agree it is not a perfect measure to evaluate one's ability to perform MI.

Response 8: We thank the reviewer for the positive comment.

Point 9: Line 210-211: This study has several limitations. First, there are other methods to evaluate motor  imagery ability, such as the mental rotation of the hand task, in addition to the motor  imagery vividness and mental chronometry tasks used in the present study

⇒Yes, I agree, EEG for example is much more specific, at least for upper-limb evaluation of Motor Imagery.

Response 9: We have taken the reviewer's valuable comments into consideration and added the text to lines 266-267.

(lines 266-267: In addition, there is a more specific method of evaluating motor imagery using EEG.)

Round 2

Reviewer 1 Report

Comments and Suggestions for Authors

Thank you for resubmitted the significantly improved your manuscript.

P2, line 86 

This sentence is the hypothesize. But, I didn't think the hypothesize.

This sentence is significantly luck of theological background in the Mirror therapy and motor imagery.

What is mechanism of Mirror therapy and motor imagery?

How relationship these mechanism?

Author should have more clearly the statement.

p3, line 124

Which is the imagery type?  Visual ? Kinesthetic?

P5, line 209

Please add the raw waveform of rest and MT condition. 

Author Response

Dear Reviewer

Thank you very much for reviewing our manuscript and offering valuable advice.
We have addressed your comments with point-by-point responses, and revised the manuscript accordingly. Please see the attachment. Thank you for your helpful recommendation.
